# Pathotyping of Newcastle Disease Virus: a Novel Single BsaHI Digestion Method of Detection and Differentiation of Avirulent Strains (Lentogenic and Mesogenic Vaccine Strains) from Virulent Virus

Perumal Arumugam Desingu,[a,b] Shambhu Dayal Singh,[a] Kuldeep Dhama,[a] Obli Rajendran Vinodhkumar,[c] K. Nagarajan,[d] Rajendra Singh,[a] Yashpal Singh Malik,[e] Raj Kumar Singh[f]

[a]Avian Diseases Section, Division of Pathology, ICAR-Indian Veterinary Research Institute, Izatnagar, Uttar Pradesh, India
[b]Department of Microbiology and Cell Biology, Indian Institute of Science, Bangalore, Karnataka, India
[c]Division of Epidemiology, ICAR-Indian Veterinary Research Institute, Izatnagar, Uttar Pradesh, India
[d]Department of Veterinary Pathology, Madras Veterinary College, Tamil Nadu Veterinary and Animal Sciences University, Vepery, Chennai, Tamil Nadu, India
[e]Division of Biological Standardization, ICAR-Indian Veterinary Research Institute, Izatnagar, Uttar Pradesh, India
[f]ICAR-Indian Veterinary Research Institute, Izatnagar, Uttar Pradesh, India

**ABSTRACT**   We provide a novel single restriction enzyme (RE; BsaHI) digestion approach for detecting distinct pathotypes of Newcastle disease virus (NDV). After scanning 4,000 F gene nucleotide sequences in the NCBI database, we discovered a single RE (BsaHI) digestion site in the cleavage site. APMV-I "F gene" class II-specific primer-based reverse transcriptase PCR was utilized to amplify a 535-bp fragment, which was then digested with the RE (BsaHI) for pathotyping avian NDV field isolates and pigeon paramyxovirus-1 isolates. The avirulent (lentogenic and mesogenic strains) produced 189- and 346-bp fragments, respectively, but the result in velogenic strains remained undigested with 535-bp fragments. In addition, 45 field NDV isolates and 8 vaccine strains were used to confirm the approach. The sequence-based analysis also agrees with the data obtained utilizing the single RE (BsaHI) digestion approach. The proposed technique has the potential to distinguish between avirulent and virulent strains in a short time span, making it valuable in NDV surveillance and monitoring research.

**IMPORTANCE** The extensive use of the NDV vaccine strain and the existence of avirulent NDV strains in wild birds makes it difficult to diagnose Newcastle Disease virus (NDV). The intracerebral pathogenicity index (ICPI) and/or sequencing-based identification, which are required to determine virulent NDV, are time-consuming, costly, difficult, and cruel techniques. We evaluated 4,000 F gene nucleotide sequences and discovered a restriction enzyme (RE; BsaHI) digestion technique for detecting NDV and vaccine pathotypes in a short time span, which is cost-effective and useful for field cases as well as for large-scale NDV monitoring and surveillance. The data acquired using the single RE BsaHI digestion technique agree with the sequence-based analysis.

**KEYWORDS** Newcastle disease, pathotyping, RT-PCR, enzyme digestion, DIVA

Address correspondence to Perumal Arumugam Desingu, perumald@iisc.ac.in, or Kuldeep Dhama, kdhama@rediffmail.com.

Newcastle disease (ND) remains one of the most complex diseases to control in commercial and backyard poultry around the world. Newcastle disease virus (NDV) is a member of the genus *Avulavirus*, which belongs to the family *Paramyxoviridae* in the order *Mononegavirales*. Avian paramyxoviruses (APMVs) are categorized into 15 serotypes (APMVs 1 to 15); with APMV-1 containing all economically relevant, naturally occurring strains of NDV (1–4). APMV-1 is separated into two clades, class I and class II,

with class II subdivided into 21 genotypes (5–8). Class I isolates have a low virulence (lentogenic) and are mostly found in waterfowls (5, 7, 9, 10), while class II isolates have high virulence and are found in poultry, pets, and wild birds (5, 7, 9). The previous five panzootic outbreaks were caused by class II viruses (7, 11–14). In addition, ND is divided into five pathotypes based on clinical signs and pathological lesions: (i) viscerotropic velogenic Newcastle disease (VVND) with digestive tract hemorrhagic lesions, (ii) neurotropic velogenic Newcastle disease (NVND) with respiratory and neurological signs, (iii) mesogenic pathotype (a less pathogenic form of NVND), (iv) lentogenic pathotype with a mild or inapparent respiratory infection, and (v) asymptomatic with no obvious disease. The intracerebral pathogenicity index (ICPI) in day-old chicks, as well as the demonstration of numerous basic amino acids at the fusion (F) protein cleavage site, are also required to report an NDV epidemic (8) (OIE, Terrestrial manual, 2021). ICPI is a time-consuming, expensive, arduous, and brutal procedure. Furthermore, due to widespread use of the NDV vaccine strain and the presence of avirulent NDV strains in wild birds, sequence analysis of the RT-PCR result is not suitable for all field instances.

Various fast techniques, such as RT-PCR followed by restriction enzyme (RE) analysis, became available with the development of molecular tools for discriminating between avirulent and virulent NDV isolates (15–18), and low-virulence lentogenic field and vaccine strains were distinguished from mesogenic and velogenic field strains by RE digestion with BglI (19). However, most of them fail to differentiate mesogenic vaccine strains from velogenic strains. The BglI restriction site-based technique is one of the best techniques, although it has resulted in lentogenic sequences being potentially misidentified as mesogenic and/or velogenic (20). The RT-PCR developed (21, 22) with specific reverse primers to differentiate NDV pathotypes has a limited ability to amplify all virulent and avirulent strains of APMV-1 (including pigeon PMV-1). Probe hybridization (23, 24) and TaqMan fluorogenic probe hybridization (25) are additional techniques to differentiate NDV pathotypes; however, these methods may not be fit for routine laboratory diagnosis with limited facilities and could possibly increase the cost and time of the diagnosis.

In the present study, for the first time, we analyzed 4,000 NDV sequences available in NCBI and identified a single RE site in the F gene cleavage site that can be used to differentiate avirulent (including lentogenic and mesogenic vaccine strains) NDV pathotypes from virulent strains. This appears to be the first report of RT-PCR-based single restriction enzyme digestion technique to differentiate avirulent, lentogenic, and mesogenic strains from virulent NDV strains for easy, economic, and specific diagnosis.

## RESULTS

**Identification of single restriction site (BsaHI) at F gene cleavage site.** Of the 4,000 NDV, F gene nucleotide, and amino acid sequences analyzed, 357 were class I-specific sequences of APMV-1 and did not possess a *BsaHI* restriction site within the F gene cleavage site. Thirteen amino acid motifs ($^{112}$G/E/V/K-K/R/Q-Q-E/D/G-R/Q-L-I/V-G-A$^{120}$) at the F gene cleavage site (see Table S1 in the supplemental material) were identified among these sequences. Motif patterns $^{115}$ERL$^{117}$, $^{115}$DRL$^{117}$, $^{115}$GRL$^{117}$, and $^{115}$EQL$^{117}$ were present in 91.32%, 4.76%, 3.64%, and 0.28% of the sequences analyzed, respectively. All patterns had an 'L' at 117 positions in the F gene cleavage site. When analyzed, the 858 class II avirulent NDV sequences showed 13 amino acid motifs ($^{112}$G/E/R-K/R/T-Q-G/K/A/R-R-L-I/L-G-A/F$^{120}$) with the patterns $^{115}$GRL$^{117}$, $^{115}$KRL$^{117}$, $^{115}$ARL$^{117}$, and $^{115}$RRL$^{117}$ present in 98.94%, 0.82%, and 0.12% of the sequences, respectively. These 13 different amino acid motif patterns have 42 (see Tables S2 and S3) different nucleic acid patterns and at least one BsaHI digestion site. The nucleotide sequence GG(A/G)CG(T/C)CT(T/G/C), corresponding to the amino acid pattern $^{115}$G/R-R-L$^{117}$ at the F gene cleavage site in class II-specific avirulent and lentogenic vaccine strain sequences in APMV-1, has a BsaHI digestion site (GGCGCC). The motif $^{115}$K/A-R-L$^{117}$, present in seven sequences (less than 0.94% of class II avirulent sequences) from India, did not have a BsaHI digestion site at the F gene cleavage site, but a $^{119}$GA$^{120}$ pattern was noticed adjacent to the F gene

cleavage site. Out of all the class II-specific avirulent NDVs, 54.19% had BsaHI digestion sites both in and adjacent to the F gene cleavage site, and the remaining 45.81% sequences had BsaHI digestion sites at only the F gene cleavage site. The analysis of 2,584 sequences of class II virulent NDV strains showed 19 different amino acid motifs ([112]R/K/G-R/R-Q/K/R/-K/R/G-R-F-I/V/L-G/S-A[120]) with the patterns [115]KRF[117], [115]RRF[117], and [115]GRF[117] in 89.97%, 9.99%, and 0.04% of the sequences analyzed, respectively. The 19 different amino acid motif patterns had 134 different nucleic acid patterns (see Table S4 and S5) with no BsaHI digestion sites.

Notably, the mesogenic strains had the same motif pattern at the F gene cleavage site as the pathogenic/velogenic class-II APMV-I strains. The mesogenic and velogenic strains also had the same amino acid motif [119]GA[120] adjacent to the cleavage site, but in mesogenic strains, the nucleotide sequence at this region corresponded to the BsaHI digestion site (GGCGCC). In velogenic strains, the nucleotide sequence at this region was found to be GG(C/T) GC(C/T) (see Tables S6 and S7) with at least one T in the degenerate nucleotide position.

**Relationship between sequence analysis and ICPI pathotyping.** The amino acid motif at the F gene cleavage site, the nucleotide pattern at F gene cleavage, and the ICPI were analyzed for 212 sequences/strains. The BsaHI digestion of these sequences showed that, in 200 sequences, the ICPI matched with the availability of RE digestion site, but 12 (12/212) sequences did not match (see Table S8).

**Relationship between sequence analyses with MDT Pathotyping.** The 82 NDV isolates/strains sequences with MDT, for which the amino acid motif at the F gene cleavage site and the nucleotide pattern at the F gene cleavage site were accessible, accurately matched when subjected to BsaHI analysis (Table S9).

**Vaccine strains.** The sequences of all NDV vaccine strains except Mukteswar and the H vaccine strain showed a BsaHI digestion site at the F gene cleavage site (Table S10).

**Validation of *in silico* analysis in laboratory conditions.** In this study, to validate the *in silico* analysis results, 45 field isolates from India and 8 commonly used vaccine (lentogenic and mesogenic) strains were used, where all the isolates and vaccine strains were found to be positive for 535-bp, class II-specific APMV-1 RT-PCR. The avirulent, lentogenic, and mesogenic strains, excluding Mukteshwar, provided digested fragments when the class II-specific RT-PCR products of 535 bp were exposed to BsaHI digestion, however, the virulent NDV RT-PCR product remained undigested by this enzyme (Fig. 1 and Table 1).

The F gene PCR products of the 44 field isolates were sequenced and submitted to GenBank (Table 1). These sequences were aligned with the 8 vaccine strain sequences to identify the F gene cleavage site/amino acid motif across sequences. In comparison of BsaHI digestion patterns across sequences with the F gene cleavage site, a 100% correlation was observed between F gene cleavage site amino acid motif-based pathotyping and BsaHI digestion pattern-based pathotyping.

## DISCUSSION

The necessity of precise and timely diagnosis of NDV outbreaks cannot be overstated since the disease is economically important and transboundary. Currently recommended methods of pathotyping of NDV include the determination of the ICPI and demonstration of the amino acid motif at the F gene cleavage site (OIE, Terrestrial manual, 2021). ICPI is time-consuming, labor-intensive, and inhumane, and has limited use in surveillance programs. Routine PCR-based diagnosis of NDV is nonconfirmatory and gives false-positive results, as many of the avirulent wild population strains and lentogenic vaccine strains used are re-isolated for several weeks post-administration of vaccine, which may give false-positive results (19, 26). For pathotypic characterization of NDV isolates, the nucleotide variation around the F gene cleavage site has been extensively exploited and is considered to be the primary molecular determinant for NDV virulence (8, 25). However, demonstration of specific amino acid motif pattern in F gene cleavage site is costly and time-consuming, requires sophisticated instruments, and has restricted use for field cases

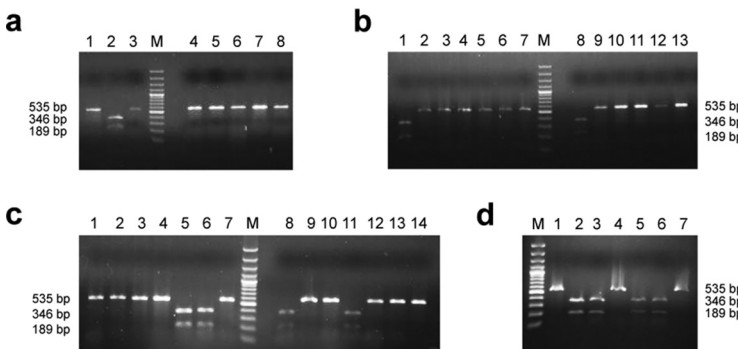

**FIG 1** NDV F gene digestion products visualized by agarose gel electrophoresis. A total of 43 isolates were tested in this study, including vaccine strains of NDV and other field isolates. All avirulent and mesogenic vaccine strains (except Mukteswar strain) of NDV were digested by BsaHI RE enzyme, while virulent strains of NDV were not digested; the agarose gel electrophoresis results are presented in panels a to d. (a) M-Ladder: lane 1, 1/Chicken/IVRI/13; lane 2, Crane/ADS/IVRI/13; lane 3, Chicken-India-UP-IVRI-0011-2010; lane 4, Emu-India-AP-IVRI-0007-2010; lane 5, Pi04/AD/94; lane 6, Chicken-India-TN-IVRI-0005-2010; lane 7, IVRI-0010; and lane 8, 129/FAIZABAD/AD-IVRI/92. (b) M-Ladder: lane 1, Hitchner B1; lane 2, 127/FAIZABAD/AD-IVRI/92; lane 3, IVRI/91; lane 4, IVRI/92; lane 5, 123/KATHUA/AD-IVRI/92; lane 6, 75/RAMPUR/AD-IVRI/89; lane 7, 108/BAREILLY/AD-IVRI/91; lane 8, 3/chicken/IVRI/13; lane 9, Pi01/AD/91; lane 10, Pi02/AD/91; lane 11, Pi05/AD/97; lane 12, Pi06/AD/01; and lane 13, PiAD388/Quail/India. (c) M-Ladder: lane 1, NDV/Peafowl/Haryana/India/IVRI-037/12; lane 2, NDV/Peafowl/UP/IVRI-024/12; lane 3, NDV/Peafowl/Delhi/IVRI-0022/12; lane 4, Mukteshwar; lane 5, Lasota; lane 6, R2B; lane 7, PiAD33/Guinea fowl/India; lane 8, Komarov; lane 9, PiAD18/Guinea fowl/India; lane 10, PiAD55/Pigeon/India; lane 11, F strain; lane 12, Chicken-India-TN-IVRI-0006-2010; lane 13, Chicken-India-TN-IVRI-0001-2010; and lane 14, Chicken-India-TN-IVRI-0003-2010. (d) M-ladder: lane 1, PiADPrd/Pigeon/India; lane 2, 2/chicken/IVRI/13; lane 3, Roakin; lane 4, Chicken-India-TN-IVRI-0002-2010; lane 5, Beaudette C; lane 6, CDF 66; and lane 7, Chicken-India-KA-IVRI-0008-2010.

and surveillance. The diagnostic methods available warrant the need for accurate, rapid, simple, and economical tests for detection and differentiation of NDV pathotypes, which is important for surveillance and control during outbreaks. In the present study on *in silico* analysis of 4,000 sequences, the RE BsaHI was identified as the RE enzyme with a cutting site either within or adjacent to the F0 cleavage site to differentiate various pathotypes. The utility of the enzyme for pathotyping was validated on 45 field isolates and 8 vaccine strains.

Class I APMV-1 includes isolates of low-virulence strains, and class II APMV-1 includes highly virulent strains, all vaccines, and reasonable numbers of avirulent strains (27). Because almost all class I NDV isolates except one, IECK90187 (10, 27) are lentogenic strains, our work relied on class II-specific detection. The most often utilized lentogenic and mesogenic vaccine strains are from class II APMV-I. From 1920 until the present, class II APMV-I viruses were responsible for all five panzootic outbreaks of NDV (12–14). The selection of the BsaHI restriction enzyme is based on the nucleotide sequences of avirulent, lentogenic, and mesogenic vaccine strains. The velogenic class-II velogenic strains did not have a BsaHI digestion site either within or adjacent to the F0 cleavage site. The class II APMV-I avirulent, lentogenic, and mesogenic vaccine strains that had a BsaHI cutting site were digested within or adjacent to the F0 cleavage site. However, the Mukteswar and H strains (vaccine strains) belong to class II genotype III of APMV-1. These strains do not contain a BsaHI RE site and have a GGTGCC nucleotide sequence at the site corresponding to the BsaHI cutting site within the F cleavage site. This indicates that these strains are velogenic strains adapted by serial egg passages to be mesogenic vaccine strains (28).

On analyzing the sequences vis-a-vis the available data on NDV pathotypes based on ICPI and MDT and comparing the amino acid motif at the cleavage site and the nucleotide pattern at the F gene cleavage site, 12 (12/278) cases were found to be mismatched between them. In any cases of mismatching between the amino acid motif at the cleavage site and pathogenicity index (ICPI), the ICPI stands the best chance of determining pathogenicity (OIE, Terrestrial manual, 2021). Although the amino acid

**TABLE 1** Comparison of F gene cleavage site amino acid motif and BsaHI digestion for detection of NDV pathotype based on 45 NDV field isolates used in this study

| Accession no. | Isolate name | Nucleotide at the F gene cleavage site | F gene cleavage site | BsaHI digestion |
|---|---|---|---|---|
| KJ627780 | 2/chicken/IVRI/13 | GGG AGA CAG GGG CGC CTT ATA GGC GCC | GRQGRLIGA | Digested |
| KJ627779 | 3/chicken/IVRI/13 | . . . . . . . . . . . . . . . . . . . . . . . . | GRQGRLIGA | Digested |
| KJ627778 | 4/chicken/IVRI/13 | . . . . . . . . . . . . . . . . . . . . . . . . | GRQGRLIGA | Digested |
| KJ627773 | Crane/ADS/IVRI/13 | . . . .A. . . . . . . . . . . . . . . . . . | GKQGRLIGA | Digested |
| KJ627783 | IVRI/92 | A.. . . . . . AAA . . . T.. . . . .T . . . | RRQKRFIGA | Not digested |
| KJ627782 | 123/KATHUA/AD-IVRI/92 | A.. . . . . . AAA . . . T.. . . . .T . . . | RRQKRFIGA | Not digested |
| KJ627781 | 1/chicken/IVRI/13 | A.. . . . . . A.A . . . T.. . . . .T . . . | RRQRRFIGA | Not digested |
| KJ627777 | 75/RAMPUR/AD-IVRI/89 | A.. . . . ..A A.A . . . T.. . . . .T . . . | RRQRRFIGA | Not digested |
| KJ627776 | 108/BAREILLY/AD-IVRI/91 | A.. . . . . . AAA . . . T.. . . . .T .G. | RRQKRFIGG | Not digested |
| KJ627775 | 127/FAIZABAD/AD-IVRI/92 | A.. . . . . . AAA . . . T.. . . . .T . . . | RRQKRFIGA | Not digested |
| KJ627774 | 129/FAIZABAD/AD-IVRI/92 | A.. . . . . . AAA . . . T.. . . . .T . . . | RRQKRFIGA | Not digested |
| KJ398401 | NDV/Peafowl/UP/IVRI-024/12 | A.. . . . . . A.A . . . T.. . . . .T . . . | RRQRRFIGA | Not digested |
| KJ398399 | NDV/Peafowl/Delhi/IVRI-0022/12 | A.. . . . . . AAA . . . T.. . . . .T . . . | RRQKRFIGA | Not digested |
| KJ627784 | IVRI/91 | A.. . . . . . AAA . . . T.. . . . .T . . . | RRQKRFIGA | Not digested |
| AJ781072 | Pi04/AD/94 | A.. . . . . . AAA . . . T.. . . . .T . . . | RRQKRFIGA | Not Digested |
| AJ781071 | Pi02/AD/91 | A.. . . . . . AAA . . . T.. . . . .T . . . | RRQKRFIGA | Not digested |
| AJ781073 | Pi05/AD/97 | A.. . . . . . AAA . . . T.. . . . .T . . . | RRQKRFIGA | Not digested |
| AJ781074 | Pi06/AD/01 | A.. . . . . . AAA . . . T.. . . . .T . . . | RRQKRFIGA | Not digested |
| AJ781075 | Pi07/AD/01 | A.. . . . . . AAA . . . T.. . . . .T . . . | RRQKRFIGA | Not digested |
| HG780860 | IVRI-0011 | A.. . . . .G. AA. . . . T.. . . . .T ..T | RRRKRFIGA | Not digested |
| HG780861 | IVRI-0007 | A.. . . . .G. AA. . . . T.. . . . .T ..T | RRRKRFIGA | Not digested |
| HG780862 | IVRI-0006 | A.. . . . .G. AA. . . . T.. . . . .T ..T | RRRKRFIGA | Not digested |
| HG780863 | IVRI-0001 | A.. . . . .G. AA. . . . T.. . . . .T ..T | RRRKRFIGA | Not digested |
| HG780864 | IVRI-0003 | A.. . . . .G. AA. . . . T.. . . . .T ..T | RRRKRFIGA | Not digested |
| HG780865 | IVRI-0002 | A.. . . . .G. AA. . . . T.. . . . .T ..T | RRRKRFIGA | Not digested |
| HG780866 | IVRI-0005 | A.. . . . .G. AA. . . . T.. . . . .T ..T | RRRKRFIGA | Not digested |
| HG780867 | IVRI-0008 | A.. . . . . . AA. . . . T.. . . . .T . . . | RRQKRFIGA | Not digested |
| HG780868 | IVRI-0009 | A.. . . . . . AA. . . . T.. . . . .T . . . | RRQKRFIGA | Not digested |
| HG780870 | IVRI-0013 | A.. . . . ..A AA. . . . T.. . . . .T . . . | RRQKRFIGA | Not digested |
| HG780869 | IVRI-0010 | A.. . . . .G. AA. . . . T.. . . . .T . . . | RRRKRFIGA | Not digested |
| AY581301 | PiAD388/Quail/India | A.. . . . . . A.A . . . T.. . . . .T . . . | RRQRRFIGA | Not digested |
| AY581302 | PiAD33/Guinea fowl/India | A.. . . . . . A.A . . . T.. . . . .T . . . | RRQRRFIGA | Not digested |
| AY581303 | PiAD18/Guinea fowl/India | A.. . . . . . A.A . . . T.. . . . .T . . . | RRQRRFIGA | Not digested |
| AY581304 | PiAD55/Pigeon/India | A.. . . . . . AAA . . . T.. . . . .T . . . | RRQKRFIGA | Not digested |
| AY581305 | PiADPrd/Pigeon/India | A.. . . . . . AAA . . . T.. . . . .T . . . | RRQKRFIGA | Not digested |
| KF750607 | Chicken-India-UP-IVRI-0011-2010 | A.. . . . .G. AA. . . . T.. . . . .T .T | RRRKRFIGA | Not digested |
| KF750608 | Chicken-India-TN-IVRI-0006-2010 | A.. . . . .G. AA. . . . T.. . . . .T .T | RRRKRFIGA | Not digested |
| KF750609 | Chicken-India-TN-IVRI-0001-2010 | A.. . . . .G. AA. . . . T.. . . . .T .T | RRRKRFIGA | Not digested |
| KF750610 | Chicken-India-TN-IVRI-0003-2010 | A.. . . . .G. AA. . . . T.. . . . .T ..T | RRRKRFIGA | Not digested |
| KF750611 | Chicken-India-TN-IVRI-0002-2010 | A.. . . . .G. AA. . . . T.. . . . .T ..T | RRRKRFIGA | Not digested |
| KF750612 | Chicken-India-TN-IVRI-0005-2010 | A.. . . . .G. AA. . . . T.. . . . .T ..T | RRRKRFIGA | Not digested |
| KF750613 | Chicken-India-KA-IVRI-0008-2010 | A.. . . . . . AA. . . . T.. . . . .T . . . | RRQKRFIGA | Not digested |
| KF750614 | Emu-India-AP-IVRI-0007-2010 | A.. . . . .G. AA. . . . T.. . . . .T ..T | RRRKRFIGA | Not digested |
| AJ781070 | Pi01/AD/91 | A.. . . . . . AAA . . . T.. . . . .T . . . | RRQKRFIGA | Not digested |

motif at the F gene cleavage site [113]RXR/KRF[117] is considered specific for virulent NDV pathotypes, the motif pattern of [112]R-R-K-K-R-F[117] in pigeon variant PMV-1 isolates has been correlated with both high- and low-virulence isolates/strains in ICPI tests conducted in chickens (29). The virulent motif [112]G-R-Q-K-R-F[117] of PPMV-1 isolates exhibited low virulence in chickens (30, 31).

There is a high similarity between virulent, vaccine, and avirulent NDV strains, which hinders the diagnosis of NDV. Recently, Liu et al. (27) demonstrated the specificity of class II-specific primers in 67 field isolates tested. In our study, all the isolates identified as class II APMV-I were confirmed to be class II of NDV using class II-specific RT-PCR (27). On sequencing 535-bp sequences amplified by RT-PCR in the 45 field isolates and comparing the F cleavage site motif pattern with the available vaccine strain sequences considered for validation, the results corroborated the findings from restriction enzyme analysis, except those for the Mukteswar vaccine strain. This novel method of

RE (BsaHI) analysis is useful for both APMV-1 and PPMV-1. Furthermore, standard sequencing is unable to distinguish between mesogenic and velogenic strains at the cleavage location, but this new procedure could differentiate mesogenic and velogenic strains. In a prior study, we used degenerate primers and nested RT-PCR to distinguish mesogenic between and avirulent NDV strains (22).

**Conclusion.** Pathotyping of NDV using RT-PCR combined with an RE digestive approach is faster and less expensive than ICPI, allowing for large-scale surveillance and detection of virulent, avirulent and vaccine strains of NDV.

## MATERIALS AND METHODS

**F gene cleavage site sequences based on *in silico* determination of restriction enzyme.** Newcastle disease virus F gene nucleotide and amino acid sequences were retrieved from GenBank NCBI (www .ncbi.nlm.nih.gov.in) and analysis of restriction enzyme sites at F gene cleavage sites was performed using NEBcutter v. 2.0 (http://tools.neb.com/NEBcutter2/) and Webcutter 2.0 (http://rna.lundberg.gu.se/cutter2/) online software.

**Use of bioinformatics tools for identification of RE.** The RE BsaHI was selected after sequence analysis of 4,000 NDV F gene sequences. On alignment, all lentogenic strains of class II APMV-1 contained a BsaHI restriction site at the F0 cleavage region. This BsaHI site was missing at the F0 cleavage region from all mesogenic and velogenic strains selected for alignment. The F gene amino acid $^{119}GA^{120}$ adjacent to the cleavage site in mesogenic vaccine strains corresponded to the BsaHI restriction site. However, no BsaHI cutting site was detected in velogenic strains.

**Validation of the technique.** Newcastle disease virus isolates (45 field isolates from India and 8 vaccine [lentogenic and mesogenic] strains) obtained from 1989 to 2013 and maintained in the Virology Laboratory Repository, Avian Disease Section, Division of Pathology, Indian Veterinary Research Institute (IVRI; Izatnagar, India) were used for this study. These isolates were reconstituted, and inoculums were prepared as explained by OIE, Terrestrial manual (2021). In brief, 0.2 mL of inoculums was inoculated into 9 to 11-day-old specific-pathogen-free (SPF) embryonated chicken eggs through the allantoic route and the eggs were incubated at 37°C until death, or for a maximum period of 120 h, whichever occurred first. The embryos were candled each day and dead embryos were chilled at 4°C overnight and collected allantoic fluids were tested for hemagglutination (HA) activity. After 120 h, all remaining live embryos were collected and chilled at 4°C overnight. The allantoic fluids found negative for HA activity were passaged further into at least one batch of eggs for confirmation. The harvested NDV was confirmed by hemagglutination inhibition test and RT-PCR.

**RNA extraction and RT-PCR.** Total RNA was extracted from allantoic fluid with TRIzol reagent (catalog no. 15596-018, Invitrogen, USA) following the manufacturer's instructions. The extracted RNA was used to synthesize cDNA using a random hexamer primer (catalog no. 15596-018, MBI Fermentas, USA). Reverse transcription (RT) for the first-strand synthesis was carried out using RevertAid H minus (M-MuLV [murine leukemia virus]-RT; MBI Fermentas, USA) in a standard 20-$\mu$L reaction mixture containing 5 $\mu$L total RNA, 1 $\mu$L random hexamer primer, 4.0 $\mu$L 5× RT buffer, 2.0 $\mu$L deoxynucleoside triphosphate mix (10 mM each), 0.5 $\mu$L RNase inhibitor and 1.0 $\mu$L M-MuLV RT enzyme (200 U/$\mu$L). Reverse transcription was carried out at 25°C for 10 min; 42°C for 60 min, and 70°C for 10 min. RT-PCR was done for all three pathotypes (lentogenic, mesogenic, and velogenic) with class II-specific fusion (F) gene primers C2-F ATGGGCYCCAGACYCTTCTAC and C2-R CTGCCACTGCTAGTTGTGATAATCC (27). The RT-PCR was optimized in a standard 25-$\mu$L reaction mixture containing 3.0 $\mu$L DNA/cDNA, 12.5 $\mu$L DreamTaq PCR Master mix 2× (Thermo Scientific, India) 1.0 $\mu$L forward primer (10 p mol/$\mu$L), 1.0 $\mu$L reverse primer (10 pmol/$\mu$L) and 7.5 $\mu$L nuclease-free water. The cyclic conditions include an initial denaturation at 95°C for 4 min, followed by 35 cycles of 94°C for 40 s, 52°C for 50 s, 72°C for 50 s, and final extension at 72°C for 5 min. The PCR products were analyzed by electrophoresis in 1.5% agarose gel stained with ethidium bromide (0.5 $\mu$g/mL).

**Restriction endonuclease analysis.** Class II-specific F gene PCR products (535 bp) were analyzed by electrophoresis. PCR products were purified using the GeneJET PCR Purification kit (Thermo Scientific, USA) according to the manufacturer's instructions. Purified PCR product was digested with BsaHI RE enzyme (New England BioLabs, United Kingdom), as per the manufacturer's instructions. The reaction included NEB buffer, 4 to 5.0 $\mu$L; BSA, 0.5 $\mu$L; PCR product, 10.0 $\mu$L (1 $\mu$g); BsaHI, 1.0 $\mu$L (10 U); and nucleus free water, 33.5 $\mu$L. The mixture was incubated at 37°C for 2 to 3 h and the reaction was stopped by storing it at −20°C until further use.

**Sequencing and sequence analysis.** Amplified F gene PCR products were purified using a GeneJET PCR purification kit (Thermo Scientific, USA) and sequenced through a commercial sequencing service (BioServe, India). The F gene nucleotide sequences were translated by DNASTAR (SeqBuilder Pro-17.2)/Expasy translation tool (https://web.expasy.org/translate/) and the corresponding amino acid motif was identified.

**Data availability.** We used nucleotide sequences related to the NDV F gene from publicly available NCBI databases in this study. Accession numbers and sequence names are provided in Table 1.

## SUPPLEMENTAL MATERIAL

Supplemental material is available online only.

**SUPPLEMENTAL FILE 1**, PDF file, 1.3 MB.

## ACKNOWLEDGMENTS

The authors are thankful to the Indian Veterinary Research Institute (Izatnagar) and the Indian Council of Agriculture Research (New Delhi) for providing the necessary facilities to carry out this research work.

P.A.D. is a DST-INSPIRE faculty member and is supported by research funding from the Government of India (DST/INSPIRE/04/2016/001067), and the Science and Engineering Research Board, Department of Science and Technology, Government of India (CRG/2018/002192).

We declare no conflicts of interest.

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
