## [Reviewer comments · Microbiology Spectrum]

Microbiology Spectrum

Pathotyping of Newcastle disease virus: A novel single BsaHI digestion method of detection and differentiation of avirulent strains (lentogenic and mesogenic vaccine strains) from virulent virus

Perumal Desingu, Shambhu Singh, Kuldeep Dhama, Obli Vinodhkumar, K Nagarajan, R. Singh, Y.S. Malik, and R. K. Singh

Corresponding Author(s): Perumal Desingu, Indian Institute of Science

Review Timeline:

Submission Date:	July 23, 2021
Editorial Decision:	September 4, 2021
Revision Received:	November 5, 2021
Accepted:	November 5, 2021

Editor: Abimbola Kolawole

Reviewer(s): The reviewers have opted to remain anonymous.

Transaction Report:

DOI: <https://doi.org/10.1128/spectrum.00989-21>

September 4, 2021

Dr. Perumal Arumugam Desingu
Indian Institute of Science
Department of Microbiology and Cell Biology
Bengaluru
India

Re: Spectrum00989-21 (Pathotyping of Newcastle disease virus: A novel single BsaHI digestion method of detection and differentiation of avirulent strains (lentogenic and mesogenic vaccine strains) from virulent virus)

Dear Dr. Perumal Arumugam Desingu:

Thank you for submitting your manuscript to Microbiology Spectrum. When submitting the revised version of your paper, please provide (1) point-by-point responses to the issues raised by the reviewers as file type "Response to Reviewers," not in your cover letter, and (2) a PDF file that indicates the changes from the original submission (by highlighting or underlining the changes) as file type "Marked Up Manuscript - For Review Only". Please use this link to submit your revised manuscript - we strongly recommend that you submit your paper within the next 60 days or reach out to me. Detailed information on submitting your revised paper are below.

Link Not Available

Sincerely,

Abimbola Kolawole

Journals Department
Reviewer comments:

Reviewer #2 (Comments for the Author):

The manuscript by Arumugam et al., describes a novel molecular approach to distinguish between distinct pathotypes of Newcastle disease virus (NDV). NDV pathotypes can be distinguished by the use of different techniques including intracerebral pathogenicity index (ICPI), RT-PCR, RT-PCR followed by restriction analysis, and probe hybridization assays. Each of these has its own limitations. In this context, the manuscript by Arumugam analyzed 4,000 NDV sequences and found a unique BsaHI restriction site in the F gene cleavage site (F0). In-silico analysis found that this site was present in all lentogenic strains of class II avian paramyxoviruses (APMV) but missing in all mesogenic and velogenic strains studied. The technique was validated using lab-grown strains of NDV, and also 45 NDV field isolates and 8 vaccines (lentogenic and mesogenic). Overall, the manuscript is well-written and easy to follow. The molecular approach proposed for pathotyping of NDV is relevant and has significance on the surveillance of NDV. I still have some comments that could improve the manuscript.

The manuscript presents a new molecular approach for pathotyping of NDV strains. In this context, it would be interesting to know if authors compared the pathotyping results obtained with the 45 NDV field isolates and 8 vaccine strains with other

molecular techniques? Could the authors comment on this?

Line 165: Did the authors find strains with more than one BsaHI restriction site?

Line 240: adopted should be modified by adapted

Line 264: even though the proposed approach is faster and cheaper compared to ICPI, it is not crystal clear that it is faster and cheaper compared to the other molecular techniques described in the literature.

Staff Comments:

Preparing Revision Guidelines

Please return the manuscript within 60 days; if you cannot complete the modification within this time period, please contact me. If you do not wish to modify the manuscript and prefer to submit it to another journal, please notify me of your decision immediately so that the manuscript may be formally withdrawn from consideration by Microbiology Spectrum.

Response to Reviewers' Comments

Reviewer #2 (Comments for the Author):

Reviewer comment: The manuscript by Arumugam et al., describes a novel molecular approach to distinguish between distinct pathotypes of Newcastle disease virus (NDV). NDV pathotypes can be distinguished by the use of different techniques including intracerebral pathogenicity index (ICPI), RT-PCR, RT-PCR followed by restriction analysis, and probe hybridization assays. Each of these has its own limitations. In this context, the manuscript by Arumugam analyzed 4,000 NDV sequences and found a unique BsaHI restriction site in the F gene cleavage site (F0). In-silico analysis found that this site was present in all lentogenic strains of class II avian paramyxoviruses (APMV) but missing in all mesogenic and velogenic strains studied. The technique was validated using lab-grown strains of NDV, and also 45 NDV field isolates and 8 vaccines (lentogenic and mesogenic). Overall, the manuscript is well-written and easy to follow. The molecular approach proposed for pathotyping of NDV is relevant and has significance on the surveillance of NDV. I still have some comments that could improve the manuscript.

Reply: Thanks very much for the clear summary and the critical comments on the work to improve the manuscript significantly.

Reviewer comment: The manuscript presents a new molecular approach for pathotyping of NDV strains. In this context, it would be interesting to know if authors compared the pathotyping results obtained with the 45 NDV field isolates and 8 vaccine strains with other molecular techniques? Could the authors comment on this?

Reply: We thank the reviewer for pointing out these to improve the manuscript. When we tested our field and vaccine strains using specific reverse primers to differentiate the NDV pathotype, these yielded results similar to those of the NDV fusion (F) protein cleavage site sequencing (Desingu *et al.*, Journal of Virological Methods 212 (2015) 47–52). Significantly, the specific reverse primers we use to differentiate the NDV pathotype have 100% nucleotide identity with our field and vaccine strains (Desingu *et al.*, Journal of Virological Methods 212 (2015) 47–52). However, in the supplementary Table 4 of this manuscript, we have shown 134 different nucleotide patterns in the fusion (F) protein cleavage site of virulent NDV Class II APMV-1. So we can not say for sure if this primer works with 134 types of nucleotide patterns. We believe that these are likely to apply to probe hybridization and TaqMan™ fluorogenic probe hybridization methods. However, there is no doubt that specific reverse primers to differentiate NDV pathotype, probe hybridization, and TaqMan™ fluorogenic probe hybridization methods can accurately determine most NDV strains' pathotypes. Therefore, we did not comment on specific reverse primers to differentiate NDV pathotypes. Further, probe hybridization and TaqMan™ fluorogenic probe hybridization techniques need specialized instruments. Therefore mentioned as follows in the manuscript: Line 86-89: *“The probe hybridization and TaqMan™ fluorogenic probe hybridization techniques to differentiate NDV pathotypes; however, these methods may not be befitting for routine laboratory diagnosis with limited facilities, and possibly increase the cost and time of the diagnosis.*

Reviewer comment: Line 165: Did the authors find strains with more than one BsaHI restriction site?

Reply: We thank the reviewer for raising a valuable point to address. The NDV class II specific fusion (F) gene primers we used could produce 535 bp amplicon. This 535 bp amplicon has a BsaHI restriction site in

two places. The first site present in the nucleotide sequence - GG(A/G)CG(T/C)CT(T/G/C), corresponding to the amino acid pattern ¹¹⁵G/R-R-L¹¹⁷ at F gene cleavage site in class II specific avirulent and lentogenic vaccine strain's sequence of APMV-1 has a BsaHI digestion site (GGCGCC). The second site, corresponding to the amino acid pattern ¹¹⁹GA¹²⁰ pattern, was noticed adjacent to the F gene cleavage site. This second site is present only in avirulent/lentogenic and mesogenic vaccine strains. Class II virulent NDV strains do not have any BsaHI restriction site in this 535 bp amplicon.

Out of all the class II specific avirulent NDV sequences we used, 54.19% have BsaHI digestion sites in and adjacent to the F gene cleavage site. The remaining 45.81% sequences showed the BsaHI digestion site at only the F gene cleavage site. Due to the proximity of these two BsaHI restriction sites, they did not cause any significant change in the BsaHI restriction digestion results. We have mentioned the information in the manuscript as follows:

Line 161-183: *“The 858 Class II avirulent NDV sequences when analyzed showed 13 amino acid motifs (112G/E/R-K/R/T-Q-G/K/A/R-R-L-I/L-G-A/F120) with the patterns 115GRL117, 115KRL117, 115ARL117, and 115RRL117 present in 98.94 %, 0.82 %, and 0.12 % of the sequences, respectively. These 13 different amino acid motif patterns have 42 (Supplementary Table 2 & 3) different nucleic acid patterns and have at least one BsaHI digestion site. The nucleotide sequence - GG(A/G)CG(T/C)CT(T/G/C), corresponding to the amino acid pattern 115G/R-R-L117 at F gene cleavage site in class II specific avirulent and lentogenic vaccine strain's sequence of APMV-1 has a BsaHI digestion site (GGCGCC). The motif 115K/A-R-L117 present in seven sequences (in less than 0.94 % of class II avirulent sequences) from India did not have a BsaHI digestion site at the F gene cleavage site, but 119GA120 pattern was noticed adjacent to the F gene cleavage site. Out of all the class II specific avirulent NDV, 54.19% have BsaHI digestion sites at both in and adjacent to the F gene cleavage site and the remaining 45.81% sequences showed BsaHI digestion site at only the F gene cleavage site. The analysis of 2584 sequences of Class II virulent NDV strains showed 19 different amino acid motifs (112R/K/G-R/R-Q/K/R/-K/R/G-R-F-I/V/L-G/S-A120) with the patterns 115KRF117, 115RRF117, and 115GRF117 in 89.97 %, 9.99 %, and 0.04 % of the sequences analyzed, respectively. The 19 different amino acid motif patterns have 134 different nucleic acid patterns (Supplementary table 4 & 5) with no BsaHI digestion site.*

It is of note that the mesogenic strains had the same motif pattern at the F gene cleavage site as the pathogenic/velogenic Class-II APMV-I. The mesogenic and velogenic strains also had the same amino acid motif 119GA120 adjacent to the cleavage site, but in mesogenic strains, the nucleotide sequence at this region corresponds to the BsaHI digestion site (GGCGCC). In velogenic strains, the nucleotide sequence at this region was found to be GG(C/T) GC(C/T) (Supplementary Table 6 & 7) with at least had one T in the degenerate nucleotide position.”

Reviewer comment: Line 240: adopted should be modified by adapted

Reply: We thank the reviewer for pointing out the mistake. In the revised manuscript, we have corrected this mistake, as per the review's suggestion.

Reviewer comment: Line 264: even though the proposed approach is faster and cheaper compared to ICPI, it is not crystal clear that it is faster and cheaper compared to the other molecular techniques described in the literature.

Reply: We thank the reviewer for the critical suggestion to improve the manuscript. As per the reviewer's suggestion, we have modified the sentence as follows in the revised manuscript. **Line 264-266:** *Pathotyping of NDV using the RT-PCR combined RE digestive approach is faster and cheaper than ICPI, allowing for large-scale surveillance and detection of virulent and avirulent as well as vaccine strains of NDV.*

November 5, 2021

Dr. Perumal Arumugam Desingu
Indian Institute of Science
Department of Microbiology and Cell Biology
Bengaluru
India

Re: Spectrum00989-21R1 (Pathotyping of Newcastle disease virus: A novel single BsaHI digestion method of detection and differentiation of avirulent strains (lentogenic and mesogenic vaccine strains) from virulent virus)

Dear Dr. Perumal Arumugam Desingu:

Your manuscript has been accepted, and I am forwarding it to the ASM Journals Department for publication. You will be notified when your proofs are ready to be viewed.

Sincerely,

Abimbola Kolawole
Editor, Microbiology Spectrum
